# CRITICAL SUBSET IDENTIFICATION: ENHANCING THE ROBUSTNESS OF 3D POINT CLOUD RECOGNITION AGAINST CORRUPTION

## ABSTRACT

Despite recent advancements in deep neural networks for point cloud recognition, real-world safety-critical applications present challenges due to unavoidable data corruption. Current models often fall short in generalizing to unforeseen distribution shifts. In this study, we harness the inherent set property of point cloud data to introduce a novel critical subset identification (CSI) method, aiming to bolster recognition robustness in the face of data corruption. Our CSI framework integrates two pivotal components: density-aware sampling (DAS) and self-entropy minimization (SEM), which cater to static and dynamic CSI, respectively. DAS ensures efficient robust anchor point sampling by factoring in local density, while SEM is employed during training to accentuate the most salient point-to-point attention. Evaluations reveal that our CSI approach yields error rates of 18.4% and 16.3% on ModelNet40-C and PointCloud-C, respectively, marking a notable improvement over state-of-the-art methods by margins of 5.2% and 4.2% on the respective benchmarks.

## 1 INTRODUCTION

Point cloud recognition, the process of identifying and categorizing objects or scenes represented as unordered point sets in 3D space, has pivotal applications in fields such as autonomous driving (Chen et al., 2017; Yue et al., 2018), medical imaging, virtual and augmented reality (Garrido et al., 2021), and 3D printing (Tashi et al., 2019). However, real-world deployment presents challenges including noise and distortion arising from sensor errors, occlusions, and missing data, which can significantly impair the accuracy and robustness of recognition models (Sun et al., 2022; Ren et al., 2022). For instance, LiDAR sensors, although essential to autonomous vehicles, are susceptible to adverse weather conditions, dirt accumulation, and hardware malfunctions. Similarly, in medical imaging, where point cloud data aids in 3D reconstructions from MRI or CT scans, the presence of artifacts, noise, and incomplete data — arising from limited resolution, patient movement, or implants — poses substantial challenges. Therefore, exploring avenues to enhance robustness against corrupted point cloud data is imperative.

To boost corruption robustness, the predominant strategy is data augmentation. Analogous to augmentation strategies applied to 2D images (Yun et al., 2019; Zhang et al., 2018), several methods are proposed to enhance training diversity and foster the learning of more robust features, including PointMixup (Chen et al., 2020), PointCutMix (Zhang et al., 2022a), RSMix (Lee et al., 2021a), and PointWOLF (Kim et al., 2021). However, as highlighted by Sun et al. (2022), different data augmentation techniques have varying degrees of effectiveness against distinct types of corruption. This is partially because data augmentation can inadvertently introduce noise and artifacts, thereby degrading the quality of the training dataset and potentially leading to overfitting and diminished generalization performance on unseen distribution shifts. Moreover, the process of augmenting data often relies on heuristic approaches, which may not always align with the underlying data distribution, thus not providing a principled manner to enhance robustness. In addition to data augmentation, Sun et al. (2022) illustrated that the robustness can also be improved from the model architecture perspective. By studying the robustness among various 3D architectures including PointNet (Charles et al., 2017), PointNet++ (Qi et al., 2017), DGCNN (Wang et al., 2019), *etc.*, they revealed that

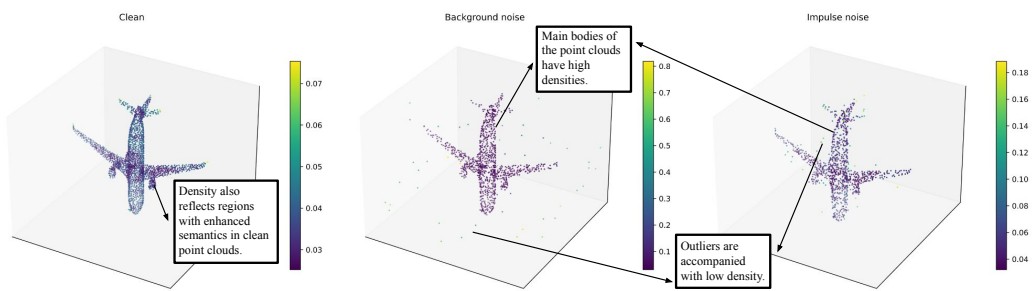

Figure 1: Illustration of Point Cloud Density. For each point, we calculate the mean distance of its $k$NN as the density score. Therefore, a smaller value denotes a larger density.

Transformers, specifically PCT (Guo et al., 2021), can significantly enhance the robustness of point cloud recognition.

In this study, we address the robustness challenge inherent in point cloud recognition by harnessing its fundamental set property, a perspective that complements existing strategies. A typical point cloud in ModelNet (Wu et al., 2015) comprises between 1000 to 2000 points. However, humans require far fewer points to recognize a point cloud accurately. While the additional points can contribute to a richer semantic understanding, they also present opportunities for data corruption. Motivated by this observation, we introduce a novel critical subset identification (CSI) method, encompassing two distinct components that facilitate both static and dynamic CSI, thereby aiding in the robust recognition of point clouds.

The first component of CSI is a novel anchor point sampling strategy to achieve static critical subset identification. Anchor point sampling is pivotal in point cloud models, facilitating local feature learning and enhancing efficiency. The strategy of Farthest point sampling (FPS) is widely adopted (Muzahid et al., 2021; Guo et al., 2021; Ma et al., 2022). While FPS is adept at preserving the skeletal structure of the given point cloud, it overlooks the potential data corruption, rendering it vulnerable to noise and distortion. We note that outlier data, typically distanced from other points, tend to exhibit lower density. Drawing from this observation, we propose a density-aware sampling (DAS) technique to down-sample the point cloud in accordance with its density. By allocating higher probabilities to points with greater density, our approach ensures a more representative subset of points is retained, thereby better preserving the semantic information inherent in the point cloud. The inherent versatility of our approach introduces an additional layer of unpredictability, fortifying the robustness of the model against data corruption.

The second component of CSI is a new optimization objective to achieve dynamic critical subset identification during training. Recent studies (Wang et al., 2021a; Zhang et al., 2022b) illustrate that employing batch-level entropy minimization in the logit space can bolster robustness during test time. Our proposition stems from the insight that entropy minimization in the embedding space could be instrumental for critical subset identification, as the objective is to enhance the saliency corresponding to the most critical point or region-level feature. By integrating the entropy minimization loss as a joint optimization objective alongside the classification objective, our method unfolds an unsupervised and architecture-agnostic solution. Through adept balancing of the entropy loss term and the classification objective, our approach cultivates enhanced robustness in confronting data corruption.

We conduct an extensive evaluation of our proposed CSI method on two widely recognized corruption robustness benchmarks: ModelNet40-C and PointCloud-C. The experimental outcomes reveal that our CSI method achieves error rates of 18.4% and 16.3% on ModelNet40-C and PointCloud-C, markedly surpassing the state-of-the-art methods by **5.2**% and **4.2**% on the respective benchmarks. Additionally, we carry out thorough ablation studies to elucidate the effectiveness of CSI from various perspectives and dimensions.

## 2 BACKGROUND AND RELATED WORK

In this section, we review a few topics that are related to our study, including deep learning models and robustness analysis for point clouds.

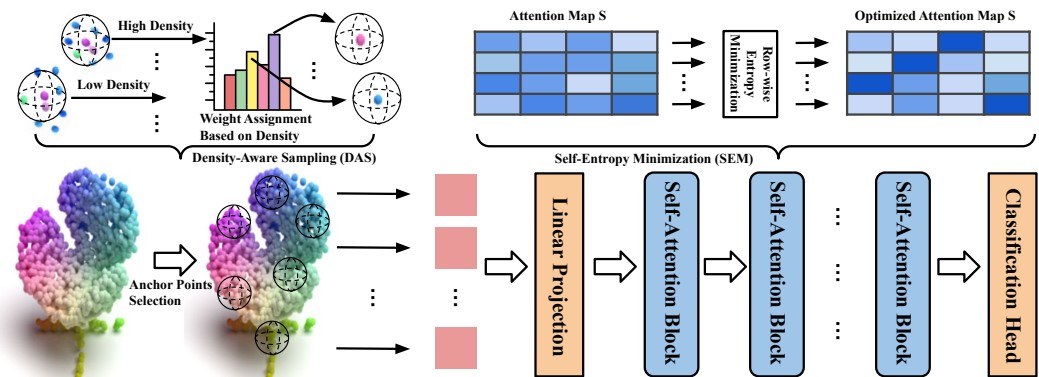

Figure 2: Overview of Our Critical Subset Identification (CSI) Method for Robust Point Cloud Recognition. CSI consists of two key components: density-aware sampling (DAS) and self-entropy minimization (SEM) objective, which achieve static and dynamic CSI, respectively. DAS ensures a robust anchor point sampling process and SEM increases the saliency of the critical correspondence between points/regions.

## 2.1 DEEP LEARNING FOR POINT CLOUD RECOGNITION

Point cloud classification is a popular research topic in scene understanding. Recent years have witnessed vigorous development of this topic owing to deep learning. The seminal work of Point-Net (Charles et al., 2017) lays the foundation by employing a multi-layer perceptron (MLP) to distill local features from each point, coupled with a max-pooling layer to amalgamate these into a global feature vector for the classification task. This further serves as a general specification for follow-up point cloud recognition studies. Building upon this, PointNet++ (Qi et al., 2017) extends the paradigm by introducing hierarchical feature learning through a series of nested local regions at varying scales. Inspired by these pioneering efforts, a slew of voxel-based methodologies such as RSCNN (Liu et al., 2019), SimpleView (Goyal et al., 2021), and RPC (Ren et al., 2022) emerge, which discretize point clouds into 3D grids of voxels, subsequently employing convolution operations on these grids to extract features. Concurrently, graph-based approaches like DGCNN (Wang et al., 2019), ECC (Simonovsky & Komodakis, 2017), SPG (Landrieu & Simonovsky, 2018), and KCNet (Shen et al., 2018) treat point clouds as graphs, leveraging graph convolutional networks to learn features. The narrative of innovation continues with the advent of attention-based methods such as PAConv (Xu et al., 2021), and Transformer-based architectures like PCT (Guo et al., 2021), Point Transformer (Zhao et al., 2021), and Point Transformer V2 (Wu et al., 2022). Transformer architectures naturally align with point cloud data processing due to their order-invariance, obviating the need for positional encoding. These methodologies harness attention mechanisms to hone in on critical regions within the point cloud, achieving state-of-the-art performance in point cloud learning, and further enriching the landscape of point cloud classification.

## 2.2 ROBUSTNESS ANALYSIS FOR POINT CLOUD RECOGNITION

Numerous efforts have been directed towards enhancing the robustness of point cloud classifiers. For instance, Triangle-Net (Xiao & Wachs, 2021) devises a feature extraction mechanism that remains invariant to positional, rotational, and scaling disturbances. While Triangle-Net exhibits remarkable robustness under severe corruption, its performance on clean data does not match the state-of-the-art benchmarks. On the other hand, PointASNL (Yan et al., 2020) introduces adaptive sampling and local-nonlocal modules to bolster robustness, albeit with a limitation of processing a fixed number of points in its implementation. Diverse strategies are explored in other works to bolster the adversarial robustness of a model. These include denoising and upsampling techniques (Wu et al., 2019), voting mechanisms on subsampled point clouds (Liu et al., 2021), exploiting the relative positions of local features (Dong et al., 2020), adversarial training (Sun et al., 2020) employing self-supervision (Sun et al., 2021), architectural improvements, and diffusion models (Sun et al., 2023). RobustPointSet (Taghanaki et al., 2020) undertakes an evaluation of point cloud classifiers' robustness under various corruption scenarios. The findings reveal that basic data augmentations

exhibit poor generalization capabilities when confronted with "unseen" corruptions. Sun et al. (2022) and Ren et al. (2022) introduced the ModelNet40-C and PointCloud-C benchmarks respectively, to facilitate a more comprehensive robustness analysis, each encompassing several types of corruption with multiple levels of severity. In this study, we utilize these benchmarks to conduct our evaluations.

# 3 CSI: A NOVEL CRITICAL SUBSET IDENTIFICATION METHOD

In this section, we introduce the design of CSI, a new critical subset identification method for point cloud recognition. As noted in § 1, point clouds fundamentally constitute a set data structure, wherein a critical subset can be sufficiently representative for recognition. Figure 2 shows the overall workflow of CSI. As shown, CSI consists of two key components: density-aware sampling (DAS) and entropy minimization objective (SEM) that achieve static and dynamic critical subset identification during model training, respectively. CSI is also applicable to most point cloud recognition architectures and we detail their formulations below.

## 3.1 DENSITY-AWARE SAMPLING

In order to improve the competence of the models to recognize fine-grained patterns and generalize to complex scenes, methods like PointNet++ (Qi et al., 2017), DGCNN (Wang et al., 2019), PCT (Guo et al., 2021) integrate neighbor embedding, trying to capture local structures via aggregating point features in the vicinity of each point. As mentioned in § 2, such neighbor embedding usually comprises both sampling and grouping steps to improve the computation efficiency. Existing studies leverage random (RS) or farthest point sampling (FPS) to select anchor points for the grouping stage. Specifically, RS, the most intuitive method, downsamples the point cloud following a uniform distribution, while FPS aims to extract the most distant point with regard to the selected point iteratively. The grouping step aggregates the features into a group, leveraging $k$ nearest neighbor ($k$NN) or ball query, to achieve local feature learning.

However, we find that both sampling methods exhibit sub-optimal performance. During the sampling phases of both training and inference stages, the inherent randomness in RS fails to ensure that the anchor points accurately represent the overarching skeleton of the object shape, resulting in diminished performance on clean inputs (detailed in § 4). Conversely, while FPS offers improved performance on clean data, it is vulnerable to exploitation through outlier points, a phenomenon frequently observed in corrupted point clouds (Sun et al., 2022; Ren et al., 2022). Motivated by these observations, we propose a simple yet effective strategy for robust anchor sampling: density-aware sampling (DAS). Our approach initially allocates a computed density weight to each point within the point cloud, followed by executing a weighted random sampling. Specifically, for each point $\boldsymbol{p}_i \in \mathbb{R}^3$, we first identify its $k$NN point set $\mathcal{S}_i$ and calculate the averaged $\ell_2$ distance between $\boldsymbol{p}_i$ and its $\mathcal{S}_i$ and further set the average distance as the threshold $t$:

$$d_i = \frac{1}{k} \sum_{j \in \mathcal{S}_i} ||\boldsymbol{p}_i - \boldsymbol{p}_j||_2, \quad t = \frac{1}{N} \sum_{i=1}^{N} d_i.$$

The sampling weight is determined by:

$$w_i = \sum_{j \in \mathcal{S}_i} \mathbf{1}[||\boldsymbol{p}_i - \boldsymbol{p}_j||_2 < t], \quad w_i = \frac{w_i}{\sum_{j=1}^{N} w_j}.$$

The underlying design philosophy of DAS is that the local density of a point usually bears a positive correlation with its significance in the point cloud, thereby facilitating a more representative sampling. This is particularly pronounced in the context of corrupted point clouds (Sun et al., 2022; Ren et al., 2022). The incorporation of weighted randomness allows for the selection of low-density points during both training and inference phases, enhancing the diversity of the sample pool with high efficiency. Moreover, the application of hard thresholding effectively eliminates extreme outliers from being considered anchors, thereby maintaining the integrity of the data.

## 3.2 SELF-ENTROPY MINIMIZATION

In this section, we explore a fresh avenue of applying the entropy minimization objective, leveraging it for dynamic critical subset identification. As underscored in § 2, Transformer architectures (Vaswani

et al., 2017) are inherently suited for 3D point cloud comprehension. We select PCT (Guo et al., 2021) as the focal architecture in our study, given its incorporation of standard self-attention modules which offer notable transferability. Furthermore, research by Sun et al. (2022) demonstrates the potential of PCT in maintaining promising performance under data corruption. Formally, given a point cloud[1] $\mathbf{P} \in \mathbb{R}^{N \times 3}$, the input embedding and sampling and grouping layers (§ 3.1) project the point cloud from the coordinate space to the feature space to obtain the embedding feature $\mathbf{F}_s \in \mathbb{R}^{M \times D}$. Then four cascaded attention layers are followed to augment the down-sampled point features. Concretely, the procedure can be expressed as below:

$$
\begin{aligned}
\mathbf{F}_1 &= \mathrm{SA}^1\left(\mathbf{F}_s\right), \quad \mathbf{F}_i = \mathrm{SA}^i\left(\mathbf{F}_{i-1}\right), \quad i = 2, 3, 4, \\
\mathbf{F}_o &= \mathrm{concat}\left(\mathbf{F}_1, \mathbf{F}_2, \mathbf{F}_3, \mathbf{F}_4\right) \cdot \mathbf{W}_o,
\end{aligned}
\tag{3}
$$

where $\mathrm{SA}^i$ represents the $i$-th self-attention layer, each having the same output dimension as its input. Specifically, let $\mathbf{Q} \in \mathbb{R}^{M \times d_a}, \mathbf{K} \in \mathbb{R}^{M \times d_a}, \mathbf{V} \in \mathbb{R}^{M \times d_e}$ be the query, key, and value matrices, respectively, generated by the linear transformation of the input features. The self-attention map $\mathbf{S} \in \mathbb{R}^{M \times M}$ and output $\mathbf{A} \in \mathbb{R}^{M \times d_e}$ are calculated by

$$
\mathbf{S} = \frac{\mathbf{Q}\mathbf{K}^\top}{\sqrt{d_a}}, \quad \mathbf{A} = \mathrm{softmax}\left(\mathbf{S}\right)\mathbf{V}
\tag{4}
$$

$\mathbf{W}_o$ is the weight of a linear layer. Before transmitting to the classification head, a max-pooling layer is applied to extract the effective global feature $\mathbf{F}_g = \mathrm{MaxPooling}\left(\mathbf{F}_o\right)$. Finally, the global feature is fed into an MLP to obtain the classification logits $\boldsymbol{x}$.

**SEM for Self-Attention Modules**. As mentioned in § 1, our goal is to highlight the critical subset within any input point clouds, which can be learned in the training stage. Entropy minimization is previously employed in test-time adaptation (Wang et al., 2021a) to enhance corruption robustness, aiming to bolster the confidence of the most prominent logit at the batch level, a strategy validated as optimal for fully test-time adaptation by Goyal et al. (2022). In this study, we propose to leverage the entropy minimization objective (SEM) for dynamic CSI during model training for robustness enhancement. The attention map, denoted as $\mathbf{S}$, is inherently aligned with our objective, encapsulating the strength of point-to-point correlations. Applying SEM to the row-wise embeddings present in $\mathbf{S}$ essentially amplifies the saliency of the most pivotal point-level feature corresponding to feature row $i$. Furthermore, row-level joint optimization can circumvent trivial solutions, fostering a more nuanced approach to achieving our goal. Specifically, our entropy minimization objective minimizes the Shannon entropy (Burks et al.) $\mathcal{H}(\mathbf{x}) = -\sum_i p_i \log p_i$ of the selected attention maps. Following Goyal et al. (2022), we utilize a temperature hyperparameter $\tau$ in SEM:

$$
\mathcal{H}\left(\mathbf{S}_{i,:}\right) = -\sum_{j=1}^{M} \frac{\exp\left(s_{i,j}/\tau\right)}{\sum_p \exp\left(s_{i,p}/\tau\right)} \log\left(\frac{\exp\left(s_{i,j}/\tau\right)}{\sum_p \exp\left(s_{i,p}/\tau\right)}\right), \quad \mathcal{L}_{\mathrm{SEM}} = \frac{1}{L}\frac{1}{M}\sum_{l=1}^{L}\sum_{i=1}^{M} \mathcal{H}\left(\mathbf{S}_{i,:}^l\right),
\tag{5}
$$

where $L$ is the amount of the specified layers applying entropy minimization loss.

**SEM for Other Models**. As outlined in § 2, the general specification for point cloud recognition involves point-level feature learning, followed by the application of a symmetric function—such as a pooling operation—to extract global features for the prediction head. Consequently, the feature map preceding this symmetric function encapsulates the most salient point-level features. We thus apply channel-wise SEM to this layer. Similarly, given a feature map $\mathbf{F} \in \mathbb{R}^{M \times d_g}$, where $M$ is the number of point-level features and $d_g$ is the number of channels, the loss term is defined as:

$$
\mathcal{H}\left(\mathbf{F}_{:,i}\right) = -\sum_{j=1}^{M} \frac{\exp\left(f_{j,i}/\tau\right)}{\sum_p \exp\left(f_{p,i}/\tau\right)} \log\left(\frac{\exp\left(f_{j,i}/\tau\right)}{\sum_p \exp\left(f_{p,i}/\tau\right)}\right), \quad \mathcal{L}_{\mathrm{SEM}} = \frac{1}{d_g}\sum_{i=1}^{d_g} \mathcal{H}\left(\mathbf{F}_{:,i}\right)
\tag{6}
$$

This objective is designed to jointly amplify the saliency of the most pivotal feature accompanied by the most critical point along each channel.

The optimization process is carried out end-to-end, integrating the cross-entropy loss $\mathcal{L}_{\mathrm{CE}}$ to formulate the loss function as $\mathcal{L} = \mathcal{L}_{\mathrm{CE}} + \lambda \mathcal{L}_{\mathrm{SEM}}$, where $\lambda$ serves as a tunable hyperparameter to harmonize the two loss components. It is also worth noting that SEM is an unsupervised task so there is no need for extra annotations.

---

[1] We use a 2D tensor with a shape of $N \times 3$ to represent the point cloud set, so permutations in the first dimension will not alter $\mathbf{P}$ in practice.

## 4 EXPERIMENTS AND RESULTS

### 4.1 EXPERIMENTAL SETUPS

**Benchmarks**. We perform our experiments on the two widely recognized benchmarks for point cloud classification, ModelNet40-C (Sun et al., 2022) and PointCloud-C (Ren et al., 2022), to evaluate the robustness of the proposed mechanisms. These benchmarks are derivatives of the foundational ModelNet40 dataset (Wu et al., 2015), each incorporating distinct types of corruption to challenge trained models under evaluation. Note that the two benchmarks are only used in test time and the training set follows the official split of the ModelNet40. Specifically, ModelNet40-C contains 15 types of common corruption, which can be categorized into density, noise, and transformation types, while PointCloud-C contains 7 types of corruption, including jittering, global/local dropping, global/local adding, scaling, and rotation. Each corruption in both datasets has 5 severity levels.

**Metrics**. Following Sun et al. (2022), we use the error rate (ER) as the metric for both ModelNet40-C and PointCloud-C. Specifically, $ER = 1 - OA$, where OA is the overall accuracy of corresponding clean/corruption data. We denote $ER_{clean}^{f}$ as the error rate for the classifier $f$ on the clean dataset (i.e., ModelNet40), $ER_{s,c}^{f}$ as the error rate for $f$ on corruption $c$ with severity $s$. The average error rate of certain corruption is defined as $ER_{c}^{f} = \frac{1}{5}\sum_{s=1}^{5} ER_{s,c}^{f}$ and the overall average error rate is $ER_{cor}^{f} = \frac{1}{C}\sum_{c=1}^{C} ER_{c}^{f}$ where $C$ is the amount of corruption types (15 for ModelNet40-C and 7 for PointCloud-C).

**Implementation Details**. As mentioned in § 3, we use the representative Transformer architecture, PCT, in our main evaluation and additionally choose nine models to demonstrate the generalization of our proposed CSI, including PointNet (Charles et al., 2017), PointNet++ (Qi et al., 2017), DGCNN (Wang et al., 2019), RSCNN (Liu et al., 2019), SimpleView (Goyal et al., 2021), GDANet (Xu et al., 2022), CurveNet (Muzahid et al., 2021), PAConv (Xu et al., 2021), and RPC (Ren et al., 2022). We adopt the same training recipe for all models for a fair comparison. Smoothed cross entropy (Wang et al., 2019) is selected as the classification loss. During training, each point cloud samples 1024 points and we use the Adam optimizer (Kingma & Ba, 2014). Each model is trained for 300 epochs and the best checkpoint is selected for evaluation. We implement our DAS based on `torch.multinomial` (Paszke et al., 2019), where the hyperparameter $k = 5$. All the experiments are performed on a single Tesla V100 GPU.

### 4.2 RESULT ON POPULAR CORRUPTION BENCHMARKS

In this section, we introduce the evaluation results of our proposed CSI on two existing corruption benchmarks: ModelNet40-C and PointCloud-C. As mentioned before, we mainly focus on PCT and we default to integrate SEM into all self-attention layers and replace FPS with DAS in the original neighbor embedding layers. On both benchmarks, we use the nine representative models with different designs described above as standalone architectures for comparisons. Details are described in the following.

**Performance on ModelNet40-C**. As shown in Table 1, our CSI-enhanced PCT achieves the best recognition performance on two corruption benchmarks, while maintaining high performance on the clean inputs. Specifically, DAS can greatly help with noise and density corruption types. We visualized the effect of DAS on certain corruptions in Figure 3, which clearly demonstrates its effectiveness. After incorporating SEM, we observe a marked performance boost, with the CSI-enhanced PCT outperforming the standard PCT on ModelNet40-C by approximately 7.1%. Additionally, we ventured into experiments with data augmentation techniques.

Table 1: Error Rates of Different Model Architectures on ModelNet40-C and PointCloud-C.

| Model (%) ↓ | ModelNet40 $ER_{clean}$ | ModelNet40-C $ER_{cor}$ | PointCloud-C $ER_{cor}$ |
|---|---|---|---|
| PointNet | 9.3 | 28.3 | 34.2 |
| PointNet++ | 7.0 | 23.6 | 24.9 |
| DGCNN | 7.4 | 25.9 | 23.6 |
| RSCNN | 7.7 | 26.2 | 26.1 |
| SimpleView | **6.1** | 27.2 | 24.3 |
| GDANet | 7.5 | 25.6 | 21.1 |
| CurveNet | 6.6 | 22.7 | 22.1 |
| PAConv | 6.4 | - | 27.0 |
| RPC | 7.0 | 26.3 | 20.5 |
| PCT | 7.1 | 25.5 | 21.9 |
| PCT+SEM | 7.2 | 21.7 | 20.9 |
| PCT+CSI | 7.3 | **18.4** | **16.3** |

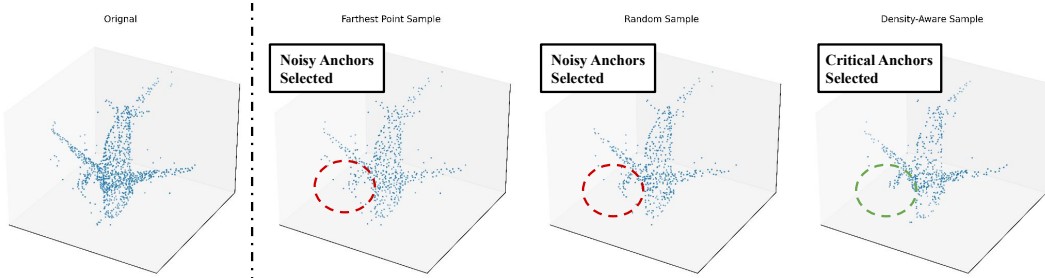

Figure 3: Example of Density-Aware Sampling (DAS). The left-most point cloud is the original one and the three sampled point clouds are selected from different sampling strategies. Our DAS effectively samples the most representative anchors statically.

Table 3: Ablation study on SEM Objective Imposed on Different Self-Attention (SA) Layers in PCT. † denotes the number of SA layers where SEM is applied.

| Model (%) ↓ | $ER_{clean}$ | $ER_{cor}$ |
|---|---|---|
| PCT | 7.1 | 25.5 |
| + SEM 1† | 7.7 | 22.0 |
| + SEM 2† | 7.9 | 22.1 |
| + SEM 3† | 7.5 | **21.5** |
| + SEM 4† | **7.2** | 21.7 |

Table 4: Ablation Study on $\lambda$ in the Overall Loss Term. SEM is applied on all self-attention (*i.e.,* 4) layers in PCT.

| $\lambda$ (%) ↓ | $ER_{clean}$ | $ER_{cor}$ |
|---|---|---|
| 1 | 9.1 | 23.5 |
| 0.5 | 8.2 | 22.1 |
| 0.25 | 7.9 | 21.9 |
| 0.1 | **7.2** | **21.7** |
| 0.05 | 7.4 | 22.2 |

Following the insights from Sun et al. (2022), who demonstrated the efficacy of PointCutMix-R in tackling data corruption, we incorporated it into our study. The results were promising; our CSI mechanism further bolstered corruption robustness, registering an improvement of 0.4%.

Table 2: Error Rates of Different Model Architectures on ModelNet40-C with Different Data Augmentation Strategies.

| Model (%) ↓ | Plain | PointCutMix-R | | |
|---|---|---|---|---|
| | $ER_{cor}$ | $ER_{cor}$ | Density | Noise | Trans. |
| PointNet | 28.3 | 21.8 | 30.5 | 18.0 | 16.9 |
| PointNet++ | 23.6 | 19.1 | 28.1 | 12.2 | 17.0 |
| DGCNN | 25.9 | 17.3 | 28.9 | 11.4 | 11.5 |
| RSCNN | 26.2 | 17.9 | 25.0 | 13.0 | 15.8 |
| SimpleView | 27.2 | 19.7 | 31.2 | 11.3 | 16.5 |
| PCT | 25.5 | 16.3 | 27.1 | 10.5 | **11.2** |
| PCT+CSI | **18.4** | **15.9** | **27.0** | **9.6** | **11.2** |

**Performance on PointCloud-C**. Table 1 also exhibits the robustness evaluation on PointCloud-C. Our CSI consistently achieves the best performance on PointCloud-C, attaining a significant improvement of 4.2% compared to the previous state-of-the-art, especially in Jitter and Add-G. Besides, we also compare with several pre-trained methods OcCo (Wang et al., 2021b) and Point-BERT (Yu et al., 2022), and data augmentation Point-Mixup (Zhang et al., 2022a), RSMix (Lee et al., 2021b), PointWOLF (Kim et al., 2021), and WOLFMix (Ren et al., 2022). Details can be found in Appendix A. The result shows that augmentation works well combined with our proposed CSI.

## 4.3 COMPREHENSIVE ABLATION STUDY

As introduced in § 3, both DAS and SEM are independent of the model architecture. In this section, we give a thorough analysis of our proposed modules. All experiments are down on ModelNet40-C.

**SEM on Different Self-Attention Layers**. In this experiment, we investigate which layers to apply our SEM objective can obtain the best gain. Specifically, we add the SEM loss in the last several layers. As shown in Table 3, imposing SEM objective in all self-attention layers accomplishes the best trade-off between $ER_{clean}$ and $ER_{cor}$.

**Weight $\lambda$ for SEM Loss**. In this experiment, we analyze the impact of our SEM weight $\lambda$ by designating a set of values {0.05, 0.1, 0.25, 0.5, 1}. We choose PCT as the baseline model and our

Table 6: Ablation Study on Different Anchor Sampling Strategies.

Table 7: Ablation study on $k$NN in Density-Aware Sampling (DAS).

| Anchor Sampling Strategy (%) ↓ | $\mathrm{ER_{clean}}$ | $\mathrm{ER_{cor}}$ |
|---|---|---|
| Farthest Point Sampling | **7.2** | 21.7 |
| Random Sampling | 7.7 | 19.6 |
| Density-Aware Sampling | 7.3 | **18.4** |

| $k$ (%) ↓ | $\mathrm{ER_{clean}}$ | $\mathrm{ER_{cor}}$ |
|---|---|---|
| 5 | **7.3** | **18.4** |
| 10 | 7.5 | 18.6 |
| 15 | 7.7 | 18.8 |
| 20 | 7.6 | 19.1 |

SEM is applied in all self-attention layers, which is our default setting. As delineated in Table 4, selecting an appropriate value for $\lambda$ is imperative to harmonize the two loss terms effectively, thereby realizing the most substantial enhancement in performance. While there exists a degree of fluctuation concerning the magnitude of improvement across different settings, the incorporation of our SEM objective continues to consistently bolster corruption robustness by a discernible margin. This consistency underscores the robustness of our SEM objective, demonstrating its capacity to contribute positively towards improving the resilience of the model against data corruption.

**SEM for Different Architectures**. In this ablation study, we extend the application of SEM across a variety of architectural designs including PointNet, PointNet++, DGCNN, RSCNN, SimpleView, and CurveNet. As elucidated in § 3.2, we formulate a generalized SEM objective tailored to the specifications of point cloud recognition. To delve deeper into its effectiveness, a comparative analysis is carried out between our method, as expressed in Equation 6, and SEM applied to the pooled global feature (i.e., $\mathbf{F}_g$ in § 3.2). The comparative results, presented in Table 5, underscore the consistent and significant enhancements our SEM objective brings about over the baseline models and other SEM objectives targeting the global feature, which is +1.34% on average. The improvement is especially pronounced in SOTA architectures. This favorable outcome substantiates our hypothesis, affirming the validity of identifying critical subsets as a robust strategy for enhancing recognition performance, especially in scenarios prone to data corruption.

Table 5: Ablation Study on SEM Imposed on Point-level (Equation 6) and Global Feature Maps with Different Architectures.

| Model (%) ↓ | $\mathrm{ER_{clean}}$ | $\mathrm{ER_{cor}}$ |
|---|---|---|
| PointNet | **9.3** | 28.3 |
| +SEM - Global Feature | 9.3 | 28.0 |
| +SEM - Point-level Feature | 9.4 | **27.1** |
| PointNet++ | 6.9 | 26.7 |
| +SEM - Global Feature | 6.9 | 26.4 |
| +SEM - Point-level Feature | **6.9** | **26.3** |
| DGCNN | 7.4 | 25.9 |
| +SEM - Global Feature | 7.3 | 25.4 |
| +SEM - Point-level Feature | **7.3** | **23.9** |
| RSCNN | **7.7** | 29.2 |
| +SEM - Global Feature | 7.7 | 26.6 |
| +SEM - Point-level Feature | 7.8 | **25.6** |
| CurveNet | 6.9 | 24.3 |
| +SEM - Global Feature | 6.9 | **22.9** |
| +SEM - Point-level Feature | **6.8** | 23.2 |

**Different Sampling Strategies**. We ablate the impacts of the sampling strategy in the neighbor embedding layers in this experiment. Our baseline model is PCT with SEM integrated. FPS serves as the sampling strategy in the original implementation of the PCT. Random Sampling (RS) can be viewed as a special case of DAS wherein all points are assumed to have equal density values. The evaluation of these strategies is encapsulated in Table 6. A key observation is that when the density of points is taken into consideration, noisy points are less likely to be sampled, which elucidates why both RS and DAS exhibit superior performance in scenarios with noise corruption. Moreover, when compared to RS, DAS demonstrates enhanced performance concerning clean inputs, thereby showcasing its potential as a more robust sampling strategy that can adeptly handle both clean and corrupted data scenarios.

$k$**NN in DAS**. § 3.1 provides an implementation outline concerning the computation of density. The choice of value $k$ in $k$NN is pivotal in determining the threshold for DAS. Generally, a larger value of $k$ corresponds to a looser threshold, which may not effectively filter out outlier noisy points, as illustrated in Table 7. After a thorough evaluation, we opt for $k = 5$ in our default DAS configuration.

**Different Definition for Density**. Apart from the density score delineated in § 3.1, we propose two alternative definitions. Firstly, density can be characterized as the median distance of all mean

Table 8: Ablation Study on Density Definition.

| Method (%) $\downarrow$ | $ER_{clean}$ | $ER_{cor}$ |
|---|---|---|
| PCT+SEM | **7.2** | 21.7 |
| + DAS - $k$NN $\ell_0$ | 7.3 | **18.4** |
| + DAS - $k$NN $\ell_1$ | 7.5 | 20.3 |
| + DAS - Ball-Query $\ell_0$ | 7.9 | 19.3 |

Table 9: Ablation Study on DAS for Different Models.

| Model (%) $\downarrow$ | $ER_{clean}$ | $ER_{cor}$ |
|---|---|---|
| CurveNet | 6.9 | 24.3 |
| + DAS | **6.6** | **23.1** |
| PointMLP | 7.1 | 32.2 |
| + DAS | **7.1** | **28.0** |

distances computed by $k$NN. The distinction between this and the approach in § 3.1 lies in the employment of the $\ell_0$ or $\ell_1$ norm. Secondly, concerning the calculation of the threshold, alternatives such as the ball query method can be employed in lieu of $k$NN. In our experimentation, we adopted the ball query method with a radius of 0.1, aligning with the implementation in Qi et al. (2017). However, after a thorough evaluation, the implementation detailed in § 3.1 was selected for its superior performance in enhancing the robustness and maintaining the clean performance.

**DAS for Different Architectures**. The concept of neighbor embedding is not confined to PCT. In this analysis, we extend our exploration to two additional models, namely CurveNet and PointMLP (Ma et al., 2022), which also incorporate neighbor embedding layers. In these methods, we substitute the FPS strategy with our DAS across all neighbor embedding layers. The outcomes of this substitution are documented in Table 9. The integration of DAS markedly enhances the robustness of these models, thereby showcasing the potential of DAS as a robust sampling strategy across different architectural frameworks.

## 5 DISCUSSION AND CONCLUSION

Robustness to data corruption is a crucial aspect in the field of 3D point cloud recognition. In real-world scenarios, the data acquired through sensors and other means often contain noise, occlusions, or other forms of corruption which can severely impede the performance of recognition systems. Unlike traditional 2D image data, 3D point clouds encapsulate geometric and spatial information which, when corrupted, can lead to a significant loss of critical information required for accurate recognition. The integrity of this data is fundamental, especially in critical applications like autonomous navigation, robotics, and real-time monitoring systems where a slight misinterpretation of the data could lead to catastrophic results. Furthermore, robustness to data corruption ensures the stability and reliability of 3D recognition systems across varying conditions and environments, making them more versatile and practical for real-world applications. Additionally, by ensuring a system's robustness to data corruption, we pave the way for more accurate and reliable models that can better generalize across different scenarios. In essence, addressing the challenges posed by data corruption in 3D point cloud recognition not only significantly enhances the performance of recognition systems but also broadens the scope and applicability of 3D recognition technologies in real-life scenarios.

In conclusion, our research leverages the unique set property of point cloud data, culminating in the development of a pioneering critical subset identification (CSI) method designed to enhance recognition robustness against data corruption. Central to our CSI approach are two key innovations: density-aware sampling (DAS) and self-entropy minimization (SEM). DAS optimizes robust anchor point sampling by considering local density, while SEM, applied during the training phase, emphasizes the most crucial point-to-point attention. Our empirical assessments underscore the efficacy of CSI, with error rates reduced to 18.4% on ModelNet40-C and 16.3% on PointCloud-C. These results represent substantial gains, outpacing SOTA methods by margins of 5.2% and 4.2% on the respective benchmarks.

**Limitations and Future Work**. Our proposed CSI demonstrates a notable improvement over standard training techniques. However, when combined with data augmentation, the margin of improvement is relatively modest. This can be attributed to the inherent complexity of data augmentation methods in 3D point clouds, which often entail the blending of point clouds from different classes. This amalgamation creates a level of ambiguity that makes the identification of a critical subset in the augmented point cloud challenging. The efficacy of DAS is primarily observed in its resilience to noise corruption. While our method offers an empirical solution to enhancing corruption robustness, a certified solution remains unexplored, presenting a potentially promising avenue for future research.

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

# A EVALUATION DETAILS

We report the detailed evaluation results in Tables 1 and 2.

Table 1: Error Rates of Different Model Architectures on ModelNet40-C and ModelNet40 with Standard Training.

| Type (%) ↓ | PointNet | PointNet++ | DGCNN | RSCNN | SimpleView | PCT | PCT+CSI |
|---|---|---|---|---|---|---|---|
| Occlusion | 52.3 | 54.7 | 59.2 | 51.8 | 55.5 | 56.6 | 56.2 |
| LiDAR | 54.9 | 66.5 | 81.0 | 68.4 | 82.2 | 76.7 | 60.2 |
| Density Inc. | 10.5 | 16.0 | 14.1 | 16.8 | 13.7 | 11.8 | 12.4 |
| Density Dec. | 11.6 | 10.0 | 17.3 | 13.2 | 17.2 | 14.3 | 12.7 |
| Cutout | 12.0 | 10.7 | 15.4 | 13.8 | 20.1 | 14.5 | 12.0 |
| Uniform | 12.4 | 20.4 | 14.6 | 24.6 | 14.5 | 12.1 | 10.8 |
| Gaussian | 14.4 | 16.4 | 16.6 | 18.3 | 14.2 | 13.9 | 10.7 |
| Impulse | 29.1 | 35.1 | 24.9 | 46.2 | 24.6 | 39.1 | 12.3 |
| Upsampling | 14.0 | 17.2 | 19.1 | 18.3 | 17.7 | 57.9 | 11.6 |
| Background | 93.6 | 18.6 | 19.1 | 29.2 | 46.8 | 18.1 | 11.2 |
| Rotation | 36.8 | 27.6 | 12.1 | 17.0 | 30.7 | 11.5 | 17.4 |
| Shear | 25.4 | 13.4 | 13.1 | 18.1 | 18.5 | 12.4 | 12.0 |
| FFD | 21.3 | 15.2 | 14.5 | 19.2 | 17.0 | 13.0 | 12.1 |
| RBF | 18.6 | 16.4 | 14.0 | 18.6 | 17.9 | 12.6 | 12.4 |
| Inv. RBF | 17.8 | 15.4 | 14.5 | 18.6 | 17.2 | 12.6 | 11.7 |
| $ER_{cor}$ | 28.3 | 23.6 | 25.9 | 26.2 | 27.2 | 25.5 | 18.4 |
| $ER_{clean}$ | 9.3 | 7.0 | 7.4 | 7.7 | 6.1 | 7.1 | 7.3 |

We also include more visualizations in Figures 1, 2, and 3 and 4 to demonstrate the effectiveness of our DAS and SEM. Our proposed DAS filters more noise data, especially under noise corruption because of the outlier point having a lower density. Also, one can observe that compared with FPS and RS, points sampled from DAS are more compact and representative.

Table 2: Full Results of the Overall Accuracy (OA). mOA: average OA over all corruptions.

| Model (%) ↑ | Clean | mOA | Scale | Jitter | Drop-G | Drop-L | Add-G | Add-L | Rotate |
|---|---|---|---|---|---|---|---|---|---|
| DGCNN | 0.926 | 0.764 | 0.906 | 0.684 | 0.752 | 0.793 | 0.705 | 0.725 | 0.785 |
| PointNet | 0.907 | 0.658 | 0.881 | 0.797 | 0.876 | 0.778 | 0.121 | 0.562 | 0.591 |
| PointNet++ | 0.930 | 0.751 | 0.918 | 0.628 | 0.841 | 0.627 | 0.819 | 0.727 | 0.698 |
| RSCNN | 0.923 | 0.739 | 0.899 | 0.630 | 0.800 | 0.686 | 0.790 | 0.683 | 0.682 |
| SimpleView | 0.939 | 0.757 | 0.918 | 0.774 | 0.692 | 0.719 | 0.710 | 0.768 | 0.717 |
| GDANet | 0.934 | 0.789 | 0.922 | 0.735 | 0.803 | 0.815 | 0.743 | 0.715 | 0.789 |
| CurveNet | 0.938 | 0.779 | 0.918 | 0.771 | 0.824 | 0.788 | 0.603 | 0.725 | 0.826 |
| PAConv | 0.936 | 0.730 | 0.915 | 0.537 | 0.752 | 0.792 | 0.680 | 0.643 | 0.792 |
| RPC | 0.930 | 0.795 | 0.921 | 0.718 | 0.878 | 0.835 | 0.726 | 0.722 | 0.768 |
| PCT | 0.930 | 0.781 | 0.918 | 0.725 | 0.869 | 0.793 | 0.770 | 0.619 | 0.776 |
| PCT+CSI | 0.927 | 0.837 | 0.895 | 0.848 | 0.861 | 0.816 | 0.905 | 0.747 | 0.787 |
| DGCNN+OcCo | 0.922 | 0.766 | 0.849 | 0.794 | 0.776 | 0.785 | 0.574 | 0.767 | 0.820 |
| Point-BERT | 0.922 | 0.693 | 0.912 | 0.602 | 0.829 | 0.762 | 0.430 | 0.604 | 0.715 |
| PN2+PointMixUp | 0.915 | 0.785 | 0.843 | 0.775 | 0.801 | 0.625 | 0.865 | 0.831 | 0.757 |
| DGCNN+PW | 0.926 | 0.809 | 0.913 | 0.727 | 0.755 | 0.819 | 0.762 | 0.790 | 0.897 |
| DGCNN+RSMix | 0.930 | 0.839 | 0.876 | 0.724 | 0.838 | 0.878 | 0.917 | 0.827 | 0.813 |
| DGCNN+WOLFMix | 0.932 | 0.871 | 0.907 | 0.774 | 0.827 | 0.881 | 0.916 | 0.886 | 0.903 |
| PointNet+WOLFMix | 0.884 | 0.743 | 0.801 | 0.850 | 0.857 | 0.776 | 0.343 | 0.807 | 0.768 |
| PCT+WOLFMix | 0.934 | 0.873 | 0.906 | 0.730 | 0.906 | 0.898 | 0.912 | 0.861 | 0.895 |
| GDANet+WOLFMix | 0.934 | 0.871 | 0.915 | 0.721 | 0.868 | 0.886 | 0.910 | 0.886 | 0.912 |
| RPC+WOLFMix | 0.933 | 0.865 | 0.905 | 0.694 | 0.895 | 0.894 | 0.902 | 0.868 | 0.897 |
| PCT+CSI+PointCutMix-R | 0.928 | 0.871 | 0.887 | 0.885 | 0.887 | 0.841 | 0.918 | 0.878 | 0.837 |

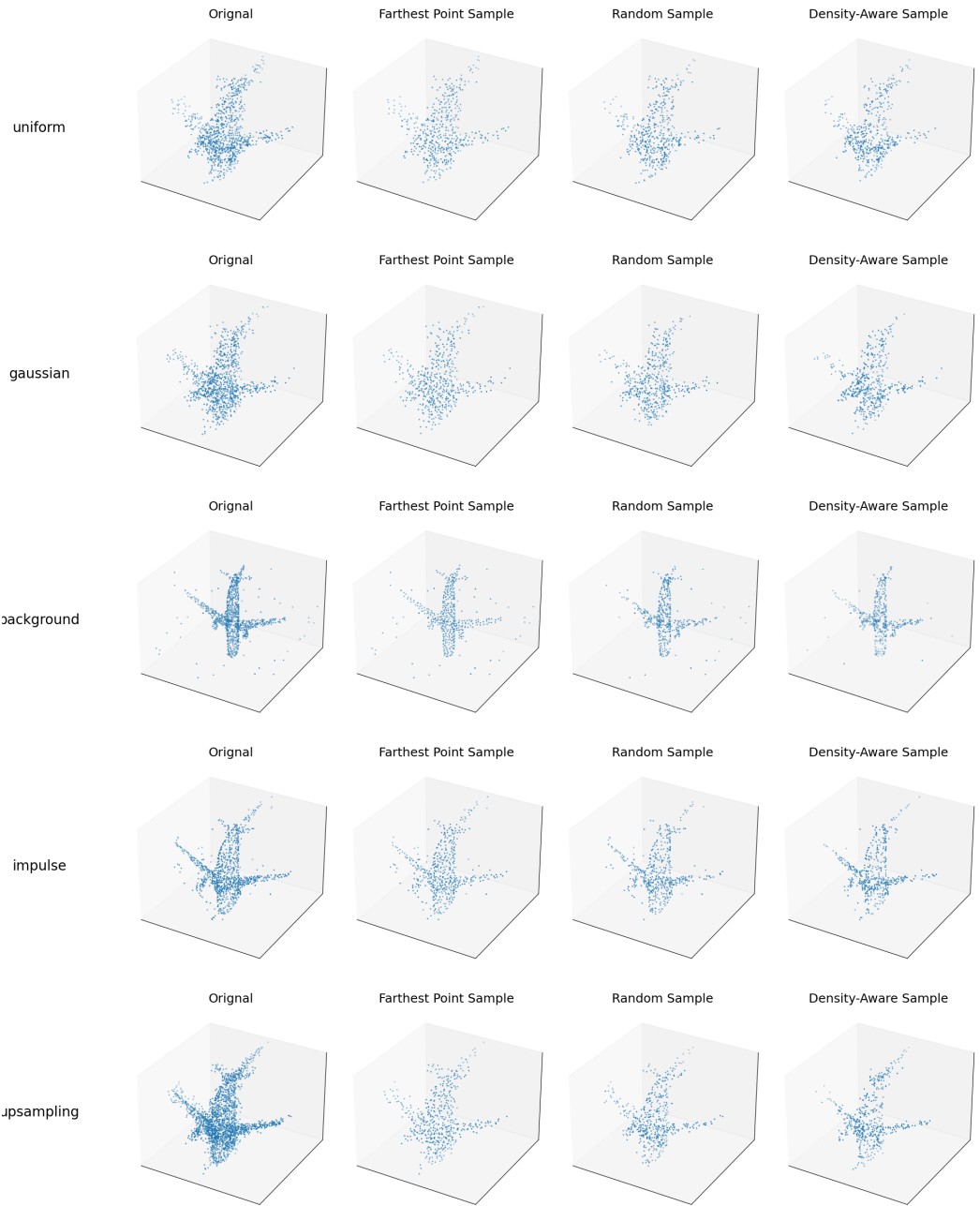

Figure 1: Visualization of different sampling strategies under noise corruption.

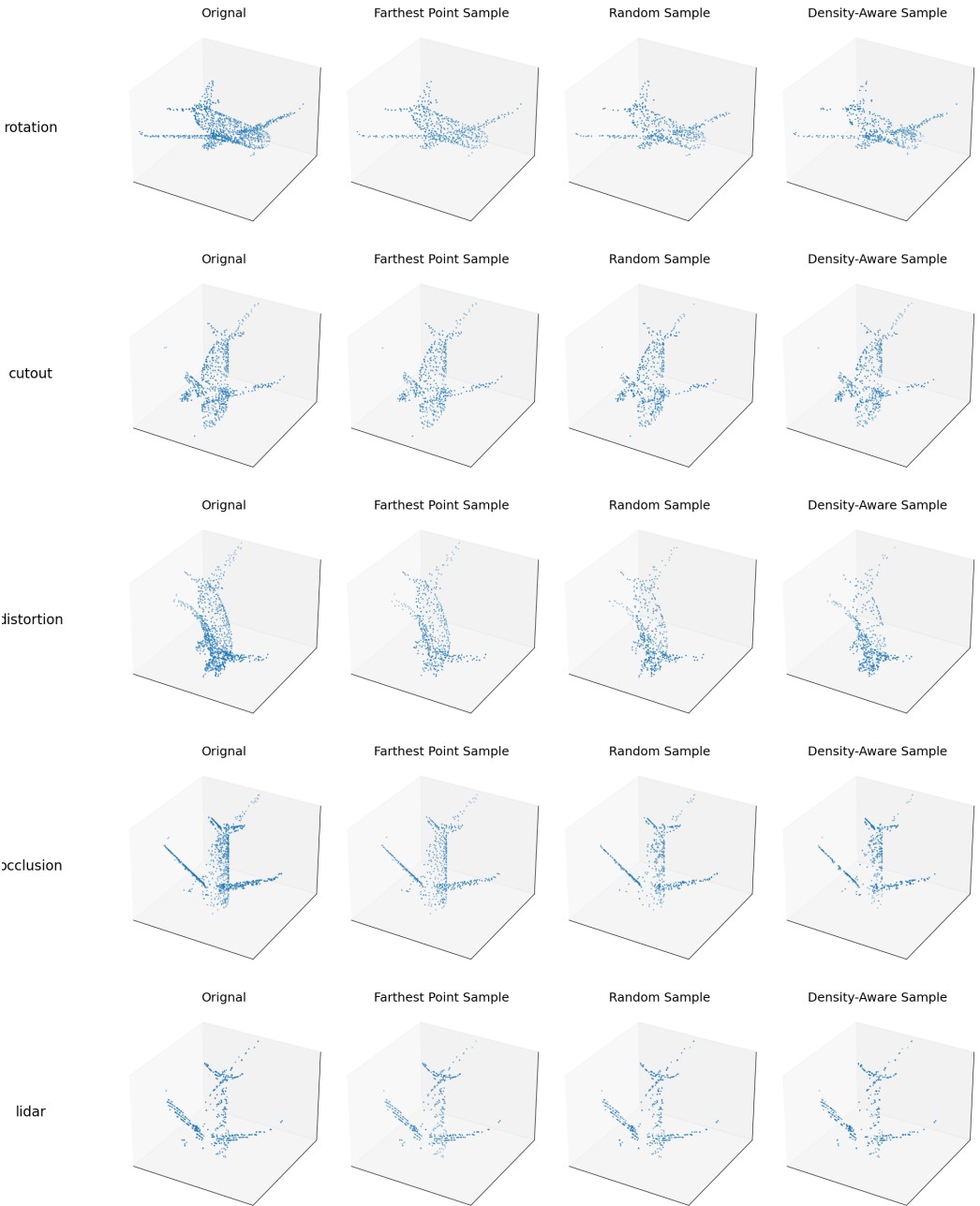

Figure 2: Visualization of different sampling strategies under the density corruption.

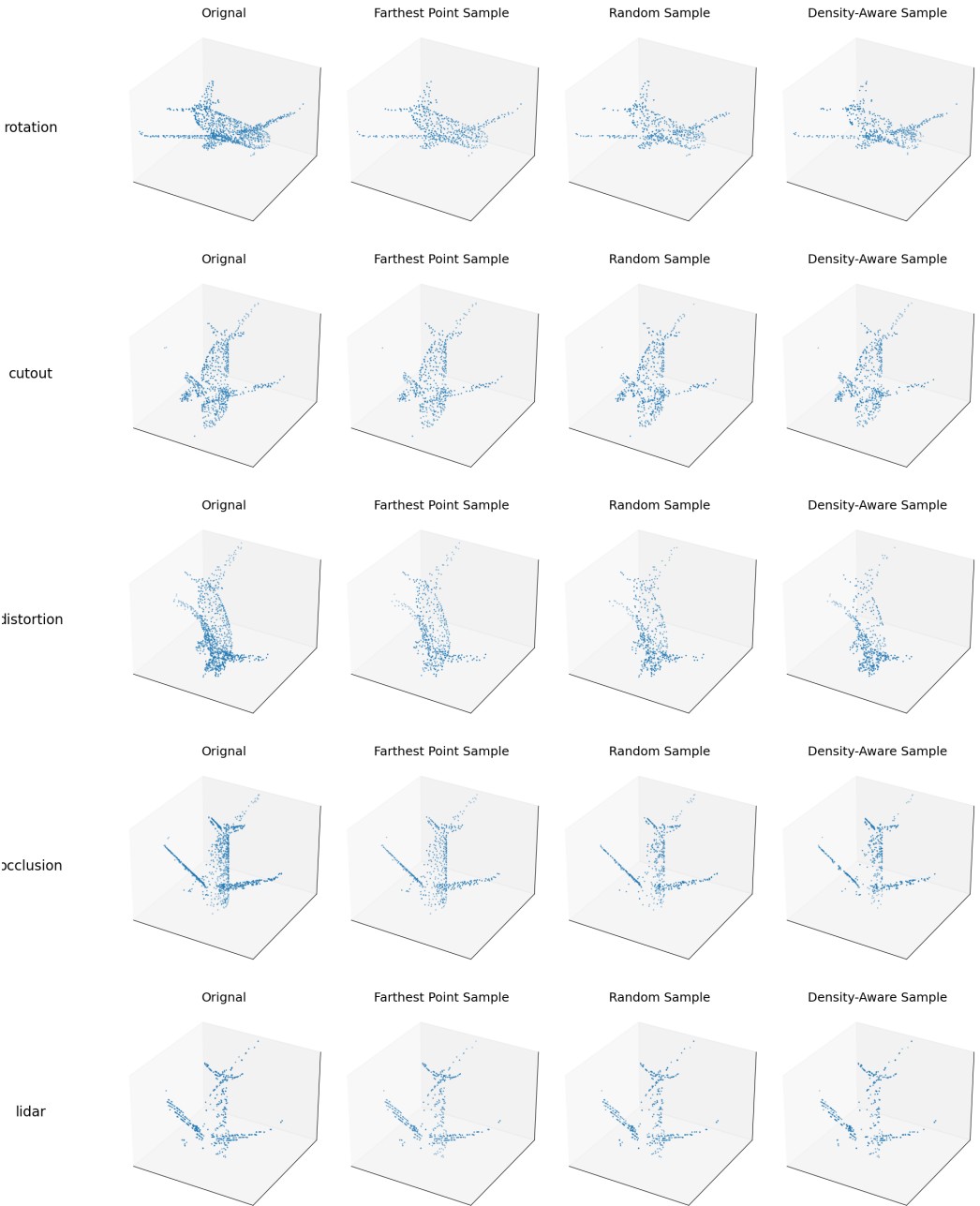

Figure 3: Visualization of different sampling strategies under the transformation corruption.

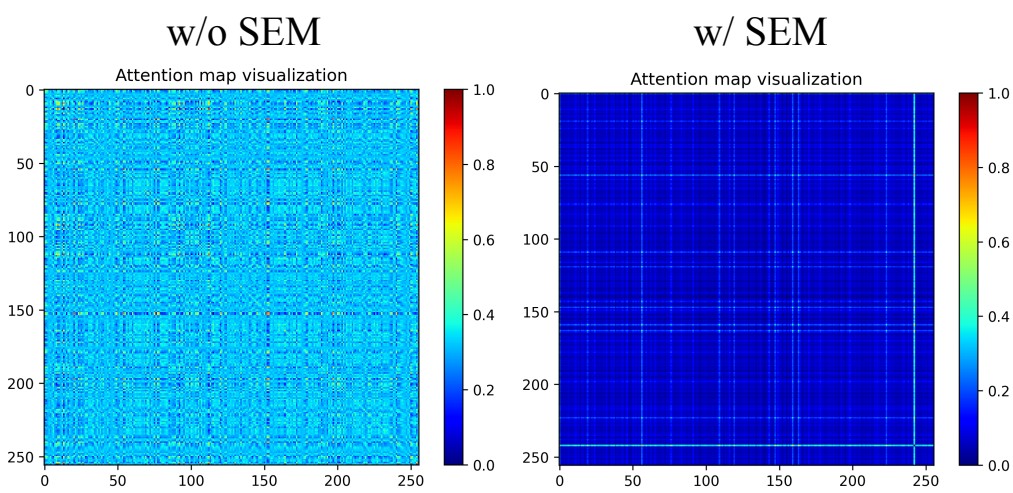

Figure 4: Attention map visualization before and after applying SEM.

