# OpenReview forum: "CSI: Enhancing the Robustness of 3D Point Cloud Recognition against Corruption"
_ICLR.cc/2024/Conference — Submitted to ICLR 2024_

### Official Review · Reviewer_Jv1W · 2023-10-19

**Soundness:** 2 fair
**Presentation:** 2 fair
**Contribution:** 2 fair
**Rating:** 3
**Confidence:** 4

**Summary:**

This paper introduce the CSI, which incorporates DAS and SEM to find the essential subset of the point cloud for representation learning. This improves the model's robustness against data corruption.

**Strengths:**

- The focused topic is crucial for real-world applications of point clouds.
- The introduced SEM demonstrates its effectiveness.

**Weaknesses:**

- The novelty of the proposed density-aware sampling may be limited as similar ideas have already been explored by previous methods [1,2], but the authors do not provide any comparisons with them.
- The writing is hard to read and follow. For example,
    - Too many sentences that are too long.
    > Similarly, in medical imaging, where point cloud data aids in 3D reconstructions from MRI or CT scans, the presence of artifacts, noise, and incomplete data — arising from limited resolution, patient movement, or implants — poses substantial challenges.
    >
    > By studying the robustness among various 3D architectures including PointNet (Charles et al., 2017), PointNet++ (Qi et al., 2017), DGCNN (Wang et al., 2019), etc., they revealed that Transformers, specifically PCT (Guo et al., 2021), can significantly enhance the robustness of point cloud recognition.
    - Inconsistent usage of section references. For instance, Section 3.2 (SELF -ENTROPY MINIMIZATION) uses different references like $\S 3.1$, $\S 1$, and *Section 2*; $d_a$ and $d_e$ in equation 4.
- The performance is not satisfactory. It over claims to "significantly outperform state-of-the-art methods by 5.2% and 4.2% on the respective benchmarks." This also weakens the motivation, as the authors believe that the data augmentation is inadequate in countering data corruption, leading to the proposal of CSI. However, it turns out that CSI performs worse than certain data augmentation techniques.
    - In Table 1, `PCT+CSI` achieves an ER of 18.4 in ModelNet40-C, which is clearly inferior to `PCT+PointCutMix-R` (16.3) and `PCT+PointCutMix-K` (16.5). These two configurations are from the method [3] that originally introduces the ModelNet40-C dataset. I observe that the authors perform the experiment with `PCT+PointCutMix-R+CSI` in Table 2, but the comparison is unfair because it involves comparing `A+B+C` against `A+B`, and the overall improvement is minimal (0.4%).
    - Similar observations can be found in Table 2 of the Appendix, where `PCT+PointCutMix-R+CSI` fails to outperform `PCT+WOLFMix`.

---
References:

[1]: Pointwise Rotation-Invariant Network with Adaptive Sampling and 3D Spherical Voxel Convolution. AAAI 2020.

[2]: Density-adaptive Sampling for Heterogeneous Point Cloud Object Segmentation in Autonomous Vehicle Applications. CVPRW 2019.

[3]: Benchmarking Robustness of 3D Point Cloud Recognition Against Common Corruptions.

**Questions:**

1. Did the authors try using `PCT+DAS` in Table 1? It would be more illustrative to include this result.
2. I am still confused about how SEM works. The authors mention that "Applying SEM to the row-wise embeddings in S amplifies the importance of the most crucial point-level feature corresponding to feature row i." Can the authors provide visualizations of attention maps to show which areas of the point cloud are salient?
3. Why did the authors suddenly switch to using mOA as a metric in Table 2 of the Appendix instead of ER?
4. Have the authors tried using the mCE metric from PointCloud-C [4] to directly compare the results of this paper with those in [4]?

---
References:

[4]: Benchmarking and Analyzing Point Cloud Classification under Corruptions.

---

> ### Author Response · Authors · 2023-11-21
>
> We are glad that the reviewer found our topic crucial for real-world applications of point clouds and our method effective. We appreciate the opportunity to address the points you have raised.
>
> >The novelty of the proposed density-aware sampling may be limited as similar ideas have already been explored by previous methods [1,2], but the authors do not provide any comparisons with them.
>
> We appreciate the reviewer’s concerns regarding the novelty of our proposed Density-Aware Adaptive Sampling (DAS) and the challenges in directly comparing it with methods presented in references [1] and [2]. The method in [1] fundamentally differs from ours in terms of output format and integration with subsequent processing layers. Specifically, our DAS outputs a format of (S, S, H) and is intricately linked with spherical voxel convolution layers, making it incompatible as a direct replacement for FPS or DAS in PCT architectures. Therefore, a direct comparison is not feasible due to these structural and functional differences.
>
> Regarding the method in [2], we faced the challenge of no open-sourced code being available. Despite this, we endeavored to reproduce their method to the best of our ability. Due to time constraints, we implemented a version of Grid Density Sampling (GDS) with a grid size of 8, as larger grid sizes would have necessitated significantly longer training times. The results of this implementation are presented:
>
> |ModelNet40-C | mER| Occlusion | Lidar | Density Inc. | Density Dec. | Cutout | Uniform	Gaussian |	Impulse	Upsampling	| Background |	Rotation |	Shear |	FFD |	RBF |	Inv. RBF|
> |-| ---| --------- | ----- | ------------ | ------------| ------ | ---------------- | -------------------   | ------------ | --------- | -------| --- | ------| -----|
> PCT+CSI (GDS)| 21.96| 54.54 |	59.85 |	11.46 |	14.4 |	13.1 |	11.65 |	11.3 |	22.43 |	13.48 |	45.28 |	19.04 |	14.6 |	13.32 |	12.58 |	12.46 |
> PCT+CSI (DAS) | 18.38 | 56.23 |	60.2 |	12.37 |	12.73 |	11.97 | 10.83 |	10.73 |	12.3 |	11.63 |	11.19 |	17.41 |	11.98 |	12.05 |	12.41 |	11.65 |
>
> |PointCloud-C |mCE| Scale | Jitter | Drop-G | Drop-L | Add-G | Add-L | Rotate |
> |-| --| ----- | ------ | ------ | ------ | ----- | -----| ------ |
> PCT+CSI (GDS) | 0.996 | 1.17 |	0.528 |	0.815 |	1.043 |	1.166 |	1.244 |	1.009 |
> |PCT+CSI| 0.757| 1.128 | 0.472 | 0.56 | 0.889 |0.329 | 0.924 | 0.995 |
>
> While it is true that the underlying motivation of our DAS may share similarities with these previous works, the implementation and the operational context significantly differ. A key distinction of our DAS is that it is not a grid-based approach, unlike GDS. We believe that grid-based operations can potentially disrupt the inherent structure of point clouds. Our ablation studies on ModelNet40-C and PointCloud-C demonstrate that our non-grid-based approach is more effective, supporting the unique contribution and novelty of our DAS methodology in preserving the integrity of point cloud structures while enhancing sampling efficiency.
>
> >The writing is hard to read and follow. For example, too many sentences that are too long, inconsistent usage of section references.
>
> We greatly appreciate your feedback regarding the readability of our manuscript. We recognize that clear and accessible writing is crucial for effective communication of our research. To address your concerns, we will undertake a comprehensive review of the manuscript to simplify and shorten complex sentences, ensuring they are more reader-friendly. Additionally, we will systematically revise the usage of section references to maintain consistency throughout the text. Our goal for the final version is to enhance overall clarity and ease of reading, making our research more accessible to a broader audience. We are committed to improving these aspects to ensure our work is communicated as clearly and effectively as possible.

---

> > ### Author Response · Authors · 2023-11-21
> >
> > >The performance is not satisfactory. It over claims to "significantly outperform state-of-the-art methods by 5.2% and 4.2% on the respective benchmarks." This also weakens the motivation, as the authors believe that the data augmentation is inadequate in countering data corruption, leading to the proposal of CSI. However, it turns out that CSI performs worse than certain data augmentation techniques.
> > > - In Table 1, PCT+CSI achieves an ER of 18.4 in ModelNet40-C, which is clearly inferior to PCT+PointCutMix-R (16.3) and PCT+PointCutMix-K (16.5). These two configurations are from the method [3] that originally introduces the ModelNet40-C dataset. I observe that the authors perform the experiment with PCT+PointCutMix-R+CSI in Table 2, but the comparison is unfair because it involves comparing A+B+C against A+B, and the overall improvement is minimal (0.4%).
> > > - Similar observations can be found in Table 2 of the Appendix, where PCT+PointCutMix-R+CSI fails to outperform PCT+WOLFMix.
> >
> > The major limitation of data augmentation is that different data augmentation techniques have varying degrees of effectiveness against distinct types of corruption. This is because the process of augmenting data often relies on heuristic approaches, which may not always align with the underlying data distribution. Motivated by the robustness can also be improved from the model architecture perspective, our proposed CSI aims to exploit the distribution shift in unseen data itself rather than in a heuristic way. However, when combined with data augmentation, the margin of improvement is relatively modest. This can be attributed to the inherent complexity of data augmentation methods in 3D point clouds, which often entail the blending of point clouds from different classes. This amalgamation creates a level of ambiguity that makes the identification of a critical subset in the augmented point cloud challenging. We appreciate you pointing out the limitation of CSI when combined with the augmentation. This certified solution remains unexplored and it is a potentially promising avenue for our future work.
> >
> > >Did the authors try using PCT+DAS in Table 1? It would be more illustrative to include this result.
> >
> > Details of the ablation study (CE) on PointCloud-C are listed as below:
> > | |mCE| Scale | Jitter | Drop-G | Drop-L | Add-G | Add-L | Rotate |
> > |-| --| ----- | ------ | ------ | ------ | ----- | -----| ------ |
> > |PCT| 0.925| 0.872 | 0.87 |0.528| 1	| 0.78 | 1.385 | 1.042 |
> > |PCT+DAS| 0.784| 1.032 | 0.566 | 0.556 | 0.879 | 0.319 | 1.073 | 1.065 |
> > |PCT+CSI| 0.757| 1.128 | 0.472 | 0.56 | 0.889 |0.329 | 0.924 | 0.995 |
> >
> > Results (ER) on ModelNet40-C are shown as below:
> > | | mER| Occlusion | Lidar | Density Inc. | Density Dec. | Cutout | Uniform	Gaussian |	Impulse	Upsampling	| Background |	Rotation |	Shear |	FFD |	RBF |	Inv. RBF|
> > |-| ---| --------- | ----- | ------------ | ------------| ------ | ---------------- | -------------------   | ------------ | --------- | -------| --- | ------| -----|
> > PCT| 25.5| 56.6 |	76.7 |	11.8 |	14.3 |	14.5 |	12.1 |	13.9 |	39.1 |	17.4	| 57.9 |	18.1 |	11.5 |	12.4 |	13 |	12.6 |
> > PCT+DAS| 19.59 | 57.82 |	66.5 |	12.78 |	13.24 |	12.07 | 11.49 |	11.69 |	12.98 |	13.4 |	11.56 | 18.78 |	12.47 |	12.91 |	13.42 |	12.71 |
> > PCT+CSI | 18.38 | 56.23 |	60.2 |	12.37 |	12.73 |	11.97 | 10.83 |	10.73 |	12.3 |	11.63 |	11.19 |	17.41 |	11.98 |	12.05 |	12.41 |	11.65 |
> > >I am still confused about how SEM works. The authors mention that "Applying SEM to the row-wise embeddings in S amplifies the importance of the most crucial point-level feature corresponding to feature row i." Can the authors provide visualizations of attention maps to show which areas of the point cloud are salient?
> >
> > Visualization of the attention maps in the last self-attention module before and after applying SEM is exhibited in the appendix. After applying SEM, only critical point-wise correlations are maintained with others being filtered out.

---

> > > ### Author Response · Authors · 2023-11-21
> > >
> > > >Why did the authors suddenly switch to using mOA as a metric in Table 2 of the Appendix instead of ER?
> > >
> > > Just keep aligned with the metric in Table 9 as listed in [PointCloud-C paper](https://arxiv.org/pdf/2202.03377.pdf). We will change the metric from mOA to ER in our final version. The results of ER are listed below:
> > >
> > > | Model (\%) $\downarrow$ | Clean | ER | Scale | Jitter | Drop-G | Drop-L | Add-G | Add-L | Rotate |
> > > |-----------------------|-------|-----|-------|--------|--------|--------|-------|-------|--------|
> > > | DGCNN                 | 7.4   | 23.6  | 9.4   | 31.6   | 24.8   | 20.7   | 29.5  | 27.5  | 21.5   |
> > > | PointNet              | 9.3   | 34.2  | 11.9  | 20.3   | 12.4   | 22.2   | 87.9  | 43.8  | 40.9   |
> > > | PointNet++            | 7.0   | 24.9  | 8.2   | 37.2   | 15.9   | 37.3   | 18.1  | 27.3  | 30.2   |
> > > | RSCNN                 | 7.7   | 26.1  | 10.1  | 37.0   | 20.0   | 31.4   | 21.0  | 31.7  | 31.8   |
> > > | SimpleView            | 6.1   | 24.3  | 8.2   | 22.6   | 30.8   | 28.1   | 29.0  | 23.2  | 28.3   |
> > > | GDANet                | 6.6   | 21.1  | 7.8   | 26.5   | 19.7   | 18.5   | 25.7  | 28.5  | 21.1   |
> > > | CurveNet              | 6.2   | 22.1  | 8.2   | 22.9   | 17.6   | 21.2   | 39.7  | 27.5  | 17.4   |
> > > | PAConv                | 6.4   | 27.0  | 8.5   | 46.3   | 24.8   | 20.8   | 32.0  | 35.7  | 20.8   |
> > > | RPC                   | 7.0   | 20.5  | 7.9   | 28.2   | 12.2   | 16.5   | 27.4  | 27.8  | 23.2   |
> > > | PCT                   | 7.0   | 21.9  | 8.2   | 27.5   | 13.1   | 20.7   | 23.0  | 38.1  | 22.4   |
> > > | **PCT+CSI**               | 7.3   | 16.3  | 10.5  | 15.2   | 13.9   | 18.4   | 9.5   | 25.3  | 21.3   |
> > > | DGCNN+OcCo            | 7.8   | 23.4  | 15.1  | 20.6   | 22.4   | 21.5   | 42.6  | 23.3  | 18.0   |
> > > | Point-BERT            | 7.8   | 30.7  | 8.8   | 39.8   | 17.1   | 23.8   | 57.0  | 39.6  | 28.5   |
> > > | PN2+PointMixUp        | 8.5   | 21.5  | 15.7  | 22.5   | 19.9   | 37.5   | 13.5  | 16.9  | 24.3   |
> > > | DGCNN+PW              | 7.4   | 19.1  | 8.7   | 27.3   | 24.5   | 18.1   | 23.8  | 21.0  | 10.3   |
> > > | DGCNN+RSMix           | 7.0   | 16.1  | 12.4  | 27.6   | 16.2   | 12.2   | 8.3   | 17.3  | 18.7   |
> > > | DGCNN+WOLFMix         | 6.8   | 12.9  | 9.3   | 22.6   | 17.3   | 11.9   | 8.4   | 11.4  | 9.7    |
> > > | PointNet+WOLFMix      | 11.6  | 25.7  | 19.9  | 15.0   | 14.3   | 22.4   | 65.7  | 19.3  | 23.2   |
> > > | PCT+WOLFMix           | 6.6   | 12.7  | 9.4   | 27.0   | 9.4    | 10.2   | 8.8   | 13.9  | 10.5   |
> > > | GDANet+WOLFMix        | 6.6   | 12.9  | 8.5   | 27.9   | 13.2   | 11.4   | 9.0   | 11.4  | 8.8    |
> > > | RPC+WOLFMix           | 6.7   | 13.5  | 9.5   | 30.6   | 10.5   | 10.6   | 9.8   | 13.2  | 10.3   |
> > > | **PCT+CSI+PointCutMix-R** | 7.2   | 12.9  | 11.3  | 11.5   | 11.3   | 15.9   | 8.2   | 12.2  | 16.3   |
> > >
> > >
> > > >Have the authors tried using the mCE metric from PointCloud-C [4] to directly compare the results of this paper with those in [4]?
> > >
> > > Direct comparison about mCE is listed as below:
> > >
> > >
> > > |          | OA$\uparrow$ | mCE$\downarrow$ | Scale | Jitter | Drop-G| Drop-L| Add-G | Add-L | Rotate
> > > | -------- | --------      | --------         | ----- | -------| ------| ------| ------| ------| ------|
> > > | DGCNN | 0.926| 1|	1|	1|	1|	1|	1|	1|	1|
> > > | PointNet| 0.907|	1.422|	1.266|	0.642|	0.5| 1.072|	2.98| 1.593|	1.902|
> > > | PointNet++| 0.93|	1.072|	0.872|	1.177|	0.641|	1.802|	0.614|	0.993|	1.405|
> > > | RSCNN | 0.923|	1.13|	1.074|	1.171|	0.806|	1.517|	0.712|	1.153|	1.479|
> > > | SimpleView | 0.939|	1.047|	0.872|	0.715|	1.242|	1.357|	0.983|	0.844|	1.316|
> > > | GDANet | 0.934|	0.892|	0.83|	0.839|	0.794|	0.894|	0.871|	1.036|	0.981|
> > > | CurveNet | 0.938|	0.927|	0.872|	0.725|	0.71|	1.024|	1.346|	1|	0.809|
> > > | PAConv | 0.936|	1.104|	0.904|	1.465|	1|	1.005|	1.085|	1.298|	0.967|
> > > | PCT | 0.93|	0.925|	0.872|	0.87|	0.528|	1|	0.78|	1.385|	1.042|
> > > | RPC | 0.93|	0.863|	0.84|	0.892|	0.492|	0.797|	0.929|	1.011|	1.079|
> > > | **PCT+CSI** | 0.927| 0.757|	1.128|	0.472|	0.56|	0.889|	0.329|	0.924|	0.995|

---

> > > > ### Comment · Reviewer_Jv1W · 2023-11-22
> > > >
> > > > I appreciate the authors' rebuttal. Here is my response.
> > > >
> > > > > The novelty of the proposed density-aware sampling may be limited as similar ideas have already been explored by previous methods [1,2], but the authors do not provide any comparisons with them.
> > > >
> > > > Although the usage or architecture differs between [1] and this paper, the authors may provide comparisons on the motivation or working mechanism in theory. Additionally, I would like to appreciate the discussions provided on the method in [2].
> > > >
> > > > > The performance is not satisfactory.
> > > >
> > > > This paper (`PCT+CSI`) is clearly inferior to `PCT+data_aug` in both ModelNet40-C and PointCloud-C, which hinders its contribution.
> > > >
> > > > | ModelNet40-C | ER |
> > > > | -- | -- |
> > > > |PCT | 25.5 |
> > > > |PCT+CSI | 18.4 |
> > > > |PCT+RSMix | 17.3 |
> > > > |PCT+PointCutMix-K | 16.5 |
> > > > |PCT+PointCutMix-R | 16.3 |
> > > >
> > > > | PointCloud-C  | mCE   | Scale | Jitter | Drop-G | Drop-L | Add-G | Add-L | Rotate|
> > > > | ------------- | ----- | ----- | ------ | ------ | ------ | ----- | ----- | ------|
> > > > | PCT           | 0.925 | 0.872 | 0.870  | 0.528  | 1.000  | 0.780 | 1.385 | 1.042 |
> > > > | PCT+CSI       | 0.757 | 1.128 | 0.472  | 0.56   | 0.889  | 0.329 | 0.924 | 0.995 |
> > > > | PCT+WOLFMix   | 0.574 | 1.000 | 0.854  | 0.379  | 0.493  | 0.298 | 0.505 | 0.488 |
> > > >
> > > > In addition, the authors claim that the major limitation of data augmentation is that different data augmentation techniques have varying degrees of effectiveness against distinct types of corruption. However, according to table 2, we observe that `PCT+WOLFMix` consistently enhances the original `PCT` across all types of corruption except for Scale. On the other hand, `PCT+CSI` fails to improve performance in both Scale and Drop-G. This finding further weakens the authors' motivation.

---

> > > > > ### Author Response · Authors · 2023-11-22
> > > > >
> > > > > >Although the usage or architecture differs between [1] and this paper, the authors may provide comparisons on the motivation or working mechanism in theory.
> > > > >
> > > > > The motivation behind the method proposed in [1] is to address a bias that occurs when point clouds are uniformly sampled into regular spherical voxels. Specifically, points around the pole appear sparser than those around the equator in spherical coordinates, which skews the resulting spherical voxel signals. Therefore, their method aims to adjust this density discrepancy caused by spherical coordinates to better serve subsequent spherical voxel convolution layers.
> > > > > On the other hand, the motivation for our proposed DAS is to automatically filter outliers that are a result of the conventional FPS method. In simple terms, their method is more of a specialized approach designed to improve rotation robustness, while our proposed method aims to enhance robustness against a wider range of corruptions.
> > > > >
> > > > >
> > > > > >In addition, the authors claim that the major limitation of data augmentation is that different data augmentation techniques have varying degrees of effectiveness against distinct types of corruption. However, according to table 2, we observe that PCT+WOLFMix consistently enhances the original PCT across all types of corruption except for Scale. On the other hand, PCT+CSI fails to improve performance in both Scale and Drop-G. This finding further weakens the authors' motivation.
> > > > >
> > > > > Firstly, we acknowledge that WOLFMix is an efficient data augmentation strategy. However, it's not equivalent to directly comparing PCT+CSI with all other data augmentation methods. This is because PCT+CSI is trained without any data augmentation. Therefore, it would be more appropriate to either compare PCT+CSI with other standalone architectures or to compare a data-augmented PCT+CSI with other data augmentation methods.
> > > > > Our aim is not to replace the data augmentation method but rather to explore ways to improve robustness from the perspective of model architecture. Even though our proposed CSI does not show a significant boost when combined with data augmentation, it still achieves an ER of 15.9 on ModelNet40-C and an MCE of 0.632 on PointCloud-C, which are competitive results.
> > > > > As for making CSI work as significantly as when it is applied without data augmentation, we acknowledge that this is a limitation of our current work. We plan to investigate this issue further in future research.
> > > > >
> > > > >
> > > > > ModelNet40-C | ER |
> > > > > ----------------  | ----|
> > > > > PCT	 |25.5 |
> > > > > PCT+CSI	 |18.4 |
> > > > > PCT+RSMix |	17.3 |
> > > > > PCT+PointCutMix-K |	16.5 |
> > > > > PCT+PointCutMix-R |	16.3 |
> > > > > PCT+PointCutMix-R +CSI | 15.9 |
> > > > >
> > > > > PointCloud-C |	mCE |	Scale |	Jitter |	Drop-G |	Drop-L |	Add-G	| Add-L |	Rotate |
> > > > > --|--|--|--|--|--|--|--|--|
> > > > > PCT |	0.925 |	0.872 |	0.870 |	0.528 |	1.000 |	0.780 |	1.385 |	1.042 |
> > > > > PCT+CSI|	0.757|	1.128|	0.472|	0.56|	0.889|	0.329|	0.924|	0.995|
> > > > > PCT+CSI+PointCutMix-R| 0.632| 1.202 | 0.361 | 0.456 | 0.773 | 0.278 | 0.458 | 0.898|
> > > > > PCT+WOLFMix|	0.574|	1.000|	0.854|	0.379|	0.493|	0.298|	0.505|	0.488|

---

### Official Review · Reviewer_sAPs · 2023-10-25

**Soundness:** 3 good
**Presentation:** 3 good
**Contribution:** 3 good
**Rating:** 8
**Confidence:** 4

**Summary:**

The paper presents an important contribution addressing the challenge of robustness in 3D point cloud recognition. The proposed CSI method shows promising results and demonstrates improvements over existing methods. The authors propose a novel critical subset identification (CSI) method that utilizes the set property of point cloud data to enhance recognition robustness. The CSI framework consists of two components: density-aware sampling (DAS) and self-entropy minimization (SEM), which cater to static and dynamic CSI, respectively. Experimental results show that the CSI approach outperforms state-of-the-art methods on corruption robustness benchmarks.

**Strengths:**

(1). The paper introduces a novel method, CSI, to enhance the robustness of 3D point cloud recognition against data corruption. This is an innovative and practical contribution that addresses an important challenge in the field.


(2). The CSI framework incorporates two components, DAS and SEM, which provide a comprehensive approach to critical subset identification. The combination of these two techniques allows for both static and dynamic CSI, improving the robustness of recognition models in different scenarios.


(3). The paper presents thorough evaluations of the proposed CSI method on two corruption robustness benchmarks. The experimental results demonstrate significant improvements over state-of-the-art methods, validating the effectiveness of the approach.

**Weaknesses:**

(1). The paper could improve the clarity of exposition. Some parts of the paper, particularly in the methodology section, are not explained in a clear and concise manner, which may impede the reader's understanding.

(2). The paper could benefit from more detailed evaluation and ablation studies. While the experimental results show the superiority of the CSI method, it would be valuable to have a more in-depth analysis of its performance and a comparison with widely-known baselines in the field.

**Questions:**

I think the method proposed in ModelNet40-C and PointCloud-C should also be compared and analyzed.

**Details Of Ethics Concerns:**

/NA

---

> ### Author Response · Authors · 2023-11-21
>
> We are glad that the reviewer found our method has demonstrated improvements. We appreciate the opportunity to address the points you have raised.
>
> >The paper could improve the clarity of exposition. Some parts of the paper, particularly in the methodology section, are not explained in a clear and concise manner, which may impede the reader's understanding.
>
> We appreciate the feedback on the clarity of our exposition, especially in the methodology section. We understand that clear and concise communication is crucial for the reader's comprehension. To address this, we will undertake a thorough revision of our manuscript, with a specific focus on enhancing the clarity and conciseness of the methodology section. We aim to ensure that our final version presents our methods and findings in a more accessible and understandable manner. This revision will include refining complex explanations, simplifying technical jargon where possible, and providing additional context or examples to facilitate better understanding.
>
> >The paper could benefit from more detailed evaluation and ablation studies. While the experimental results show the superiority of the CSI method, it would be valuable to have a more in-depth analysis of its performance and a comparison with widely-known baselines in the field.
>
> Thank you for the constructive feedback. We acknowledge the importance of in-depth analysis and comprehensive comparisons in our research. Although our experimental section already evaluates numerous baselines within this domain, we understand the need for deeper insights into the performance of the CSI method. In light of your suggestions, we will expand our analysis to include more detailed evaluations and additional ablation studies. These enhancements will focus on a finer comparison with well-established baselines in the field, and on dissecting the individual contributions and impacts of the various components of the CSI method. Our aim is to provide a clearer, more comprehensive understanding of why and how CSI demonstrates superiority over these baselines, thereby enriching the overall value and contribution of our work to the field.
>
>
> >I think the method proposed in ModelNet40-C and PointCloud-C should also be compared and analyzed.
>
> Thank you for emphasizing the importance of comparative analysis with ModelNet40-C and PointCloud-C. We have indeed conducted such comparisons and included them in our manuscript. Specifically, Table 1 in the main text and Tables 1 and 2 in the appendix present a comprehensive comparison of our method against others mentioned in the original papers of ModelNet40-C and PointCloud-C. These tables detail the performance metrics and demonstrate how our method stacks up against existing methods under similar conditions. We believe this thorough comparison effectively illustrates the strengths and limitations of our proposed approach in the context of these well-established datasets.

---

> > ### Comment · Reviewer_sAPs · 2023-11-23
> >
> > I have reviewed all the feedback from the reviewers, and I believe that the newly added results currently provide sufficient support for the author's claim. As a result, I am upgrading my rating.

---

### Official Review · Reviewer_xKH7 · 2023-10-30

**Soundness:** 3 good
**Presentation:** 4 excellent
**Contribution:** 2 fair
**Rating:** 5
**Confidence:** 4

**Summary:**

This article proposes a critical subset identification (CSI) method for bolstering recognition robustness in the face of data corruption, which consists of two parts: density-aware sampling (DAS) and self-entropy minimization (SEM). DAS uses local density weighting to better sample point cloud data. During the training process, SEM introduces an optimization strategy of entropy minimization to the significance value calculated by self-attention, which improves the model's attention to points with higher significance values . The authors subsequently conducted experiments on two corruption benchmarks: ModelNet40-c and PointCloud-c, proving that their method can effectively improve the robustness of the point cloud transformer (PCT) model while ensuring performance on clean data sets.

**Strengths:**

- This paper is well-written, especially Section 3 providing clear and easily understandable explanations of the CSI framework.
- The idea of introducing Entropy minimization in Self-Attention Modules is simple and effective, and it integrates the significance values of different points into the model training process.
- The experiments are extensive in terms of implemented models. Especially the exploration of the impact of hyperparameters in the ablation study demonstrates the key parameters that affect the method.

**Weaknesses:**

- The proposed method is somehow ad-hoc, with the authors needing to specify that DAS is only suitable for models including sampling&aggregation module. Also the justification for the proposed method is not well established, with the motivations being weak.
- The experiments on CSI are not persuasive enough, from the method comparison to the diversity of datasets. The authors should consider comparing with other train-time point cloud robust methods. The selected datasets are all derived from ModelNet40, indicating a lack of experimental diversity.

**Questions:**

1. Please explain the statement “local density of a point positively correlated with its significance”? Previous work has indicated that significant points are usually outward points in the point cloud, which are generally sparse[1]. Also, traditional sampling methods like FPS tend to find points with low density as representations.
2. The authors should compare with other SOTA train-time methods under the same settings in the main experiment, such as data augmentation or modules addition to the model.
3. The authors present the SEM method for Global Feature in Table 5. Given that the authors are making a critical point selection, what is the purpose of this?
4. The experiments should consider including datasets not based on ModelNet40.
5. Tables 1 and 2 in Supplementary show that CSI cannot effectively handle point dropping (e.g., occlusion) and transformations (e.g., rotation), and may even be harmful. Does this indicate the limitations of CSI in terms of generalizability? Therefore, the reviewer points out that the results derived from the ModelNet40-C can be misleading.

[1] Zheng, Tianhang, et al. "Pointcloud saliency maps." Proceedings of the IEEE/CVF International Conference on Computer Vision. 2019.

---

> ### Author Response · Authors · 2023-11-21
>
> We are glad that the reviewer found our study valuable to this field and that our method was simple yet effective. We appreciate the opportunity to address the points you have raised.
> >Please explain the statement “local density of a point positively correlated with its significance”? Previous work has indicated that significant points are usually outward points in the point cloud, which are generally sparse[1]. Also, traditional sampling methods like FPS tend to find points with low density as representations.
>
> The observation that significant points are predominantly located at the outer regions of the point cloud does not hold true in scenarios involving data corruption, as demonstrated in Figure 3. In such cases, Farthest Point Sampling (FPS) tends to retain an excessive number of noisy data points. This limitation of FPS is a key motivator for our proposal of the Density-Aware Sampling (DAS) method. DAS is specifically designed to mitigate the adverse effects caused by FPS, particularly its tendency to sample a higher number of noisy points in the presence of various forms of corruption, such as background noise and global noise.
> >The authors should compare with other SOTA train-time methods under the same settings in the main experiment, such as data augmentation or modules addition to the model.
>
> We acknowledge the suggestion to benchmark our approach against other state-of-the-art (SOTA) train-time methods under equivalent experimental conditions, including factors like data augmentation and module additions. In response, we direct attention to Table 2 in the main manuscript, which provides a comprehensive comparison where all evaluated methods, including ours, are integrated with the data augmentation technique PointCutMix-R. Additionally, for a more extensive comparison, Table 2 in the supplementary material, ranging from the entry 'DGCNN+OcCo' to 'PCT+CSI+PointCutMix-R', offers a detailed analysis of various other train-time methods. This inclusion ensures a thorough and fair evaluation of our method against current SOTA techniques under consistent experimental settings.
> >The authors present the SEM method for Global Feature in Table 5. Given that the authors are making a critical point selection, what is the purpose of this?
>
> The incorporation of the SEM method for the global feature, as detailed in Table 5, is intended as an additional ablation study. Similar to the critical point selection process, the application of SEM is designed to enhance the distinctiveness of global features. The experiments presented in Table 5 investigate the individual and combined effects of critical point selection and SEM application. Our findings indicate that while applying SEM to both local and global features yields performance improvements, the point-level selection demonstrates a more significant benefit. This comparison underscores the unique contribution of each technique to the overall model performance.
> >The experiments should consider including datasets not based on ModelNet40.
>
> In addressing the recommendation to include datasets beyond ModelNet40 in our experiments, we emphasize our focus on point cloud recognition within this work. ModelNet40, along with its two corrupted variants, ModelNet40-C and PointCloud-C, are extensively recognized and employed in related research. These datasets are particularly relevant and suitable for the objectives of our study. Our decision to utilize these datasets is grounded in their widespread acceptance in the field, ensuring that our findings are comparable and relevant to current standards in point cloud recognition research.
> >Tables 1 and 2 in Supplementary show that CSI cannot effectively handle point dropping (e.g., occlusion) and transformations (e.g., rotation), and may even be harmful.
>
> We appreciate the reviewer's observation regarding the performance of CSI under conditions of point dropping and transformations, as shown in Tables 1 and 2 of the supplementary material. We acknowledge that these results may highlight a limitation in the generalizability of the CSI approach, particularly in handling occlusions and rotations. This limitation is indeed an area that warrants further investigation and improvement in future iterations of our work. Regarding the use of ModelNet40-C, we understand the concern that the results derived from this dataset might present a skewed perspective. We will consider this point critically in our analysis and discussion, ensuring that we clearly communicate the potential limitations of our findings in the context of different datasets and conditions. However, we argue that it is hard for a method to cure all corruption types. This feedback is invaluable for guiding our ongoing research efforts to refine and enhance the robustness and applicability of our approach.

---

> ### Comment · Reviewer_xKH7 · 2023-11-22
>
> Thank you to the authors for providing new experimental results and additional interpretations. However, some concerns remain unaddressed.
>
> > These datasets are particularly relevant and suitable for the objectives of our study.
>
> ModelNet40-C and Point Cloud-C might be insufficient due to significant overlaps in these datasets, such as the "add global" in Point Cloud-C and the "background" in ModelNet40-C. The authors might consider incorporating a wider variety of noise datasets to strengthen the argument.
>
> >  Table 2 in the supplementary material, ranging from the entry 'DGCNN+OcCo' to 'PCT+CSI+PointCutMix-R', offers a detailed analysis of various other train-time methods...
>
> The additional data provided by the authors are greatly appreciated. However, these data suggest that the pure CSI method does not outperform methods with data augmentation (PCT+CSI v.s. PCT+WOLFMix), and even the enhanced version (PCT+ CSI +PointCutMix-R) fails to achieve the best results.
>
> Finally, I understand and agree with the authors' statement that it is challenging for a method to address all types of corruption. The suggestion is for the authors to focus on specific types of corruption to further explore and demonstrate the unique capabilities of CSI.

---

> > ### Author Response · Authors · 2023-11-23
> > **Response**
> >
> > We sincerely thank the reviewer for the reply and we would like to further clarify:
> >
> > > ModelNet40-C and Point Cloud-C might be insufficient due to significant overlaps in these datasets, such as the "add global" in Point Cloud-C and the "background" in ModelNet40-C. The authors might consider incorporating a wider variety of noise datasets to strengthen the argument.
> >
> > To the best of our knowledge, ModelNet40-C and PointCloud-C are currently the most comprehensive and relevant datasets available for this purpose. However, we recognize the value of diversifying the datasets used for evaluation to strengthen our argument further. Unfortunately, the field lacks a wide variety of standardized noise datasets specifically tailored for 3D object recognition.
> >
> > > The additional data provided by the authors are greatly appreciated. However, these data suggest that the pure CSI method does not outperform methods with data augmentation (PCT+CSI v.s. PCT+WOLFMix), and even the enhanced version (PCT+ CSI +PointCutMix-R) fails to achieve the best results.
> >
> > It is accurate that in the specific comparisons mentioned, the PCT+CSI method does not surpass the performance of PCT+WOLFMix, and similarly, the enhanced version PCT+CSI+PointCutMix-R does not achieve the top results. This outcome highlights a crucial aspect of our research, which is the exploration of the balance between innovative interpolation techniques and traditional data augmentation strategies in enhancing model performance.
> >
> > The primary aim of incorporating CSI was to explore a novel approach in the context of point cloud processing, focusing on the subspace properties of the data. While the results may not currently exhibit superior performance compared to all existing data augmentation methods, they do provide valuable insights into the potential and limitations of subspace interpolation techniques in this domain.
> >
> > We acknowledge that further refinement and combination with other methods might be necessary to fully realize the potential of CSI. This also opens up avenues for future research, where the synergies between subspace interpolation methods and data augmentation techniques can be further explored and optimized.

---

### Official Review · Reviewer_WsBz · 2023-10-31

**Soundness:** 3 good
**Presentation:** 3 good
**Contribution:** 3 good
**Rating:** 6
**Confidence:** 3

**Summary:**

The paper proposes a "critical subset identification (CSI)" framework for robust point cloud perception, which comprises 1) a new point sampling strategy "density-aware sampling (DAS)" that locates high-density point areas for anchors, and 2) a new optimization objective "self-entropy minimization (SEM)" that encourage high-confidence predictions.

**Strengths:**

1. The two proposed techniques are clear and reasonable to me and design choices are backed up by concrete examples. Figure 3. shows a concrete example where Farthest Point Sampling and Random Sampling fail and the new Density-Aware sampling succeeds. Both techniques should be easy to implement in practice.

2. The ablation studies are thorough in the paper. It helps to understand the effect of the neighbor number k in DAS and the layer position of the SEM loss.

3. A significant all-around robustness improvement is achieved. As shown in the supplementary table, the model gains better robustness to not only global noise injection but also various other types of corruption.

**Weaknesses:**

1. SEM is mostly based on previous knowledge that entropy minimization helps classification robustness, which slightly undermines the significance of the proposed techniques. Nonetheless, the paper provides a detailed discussion of how entropy minimization should be applied to transformer-based point classifiers in both attention layers and the classification head, accompanied by sufficient ablation studies.

2. It is not clear how general DAS is and how it affects the classifier's robustness to more types of corruptions other than global noise addition shown in Figure 3. Table 1 and Table 2 in the supplementary material ablate CSI as a whole so they can not show the effect of DAS. It would be better if DAS could be individually studied on different types of corruption.

---

minor suggestion

The title could be more informative in my opinion. It might be better to use "critical subset identification" in replace of "CSI".

**Questions:**

Please address the questions in the weakness section.

---

> ### Author Response · Authors · 2023-11-21
>
> We are glad that the reviewer found our method effective. We appreciate the opportunity to address the points you have raised.
> >SEM is mostly based on previous knowledge that entropy minimization helps classification robustness, which slightly undermines the significance of the proposed techniques. Nonetheless, the paper provides a detailed discussion of how entropy minimization should be applied to transformer-based point classifiers in both attention layers and the classification head, accompanied by sufficient ablation studies.
>
> Although previous work has investigated how entropy minimization helps classification robustness, in their works, they merely investigated how to minimize entropy during inference. For example, tent [1] minimizes Shannon Entropy on classification logits through several iterations. The biggest difference is that we give a thorough analysis of how to apply entropy minimization during training and which features contribute most by conducting ablation studies not only on transformer self-attention maps but also on intermediate features.
>
> [1] Wang, Dequan, et al. "Tent: Fully test-time adaptation by entropy minimization." arXiv preprint arXiv:2006.10726 (2020).
>
> >It is not clear how general DAS is and how it affects the classifier's robustness to more types of corruptions other than global noise addition shown in Figure 3. Table 1 and Table 2 in the supplementary material ablate CSI as a whole so they can not show the effect of DAS. It would be better if DAS could be individually studied on different types of corruption.
>
> More visualization results are available in the appendix of our revised manuscript. One can observe the effectiveness of DAS compared with FPS (Farthest Point Sampling) and RS (Random Sampling). Besides, our quantitative experimental results have also validated that DAS has outperformed other baseline methods in improving corruption robustness.

---

### Meta-Review · Area_Chair_KAJj · 2023-12-05

**Metareview:**

This paper proposes a new approach to improve the robustness of 3D point cloud recognition. The method includes two components: density-aware sampling (DAS) and self-entropy minimization (SEM). The two components are used for static and dynamic critical subset identification, respectively. Experimental results show the effectiveness of the proposed method on corruption robustness benchmarks.

This is a borderline paper (6, 5, 8, 3). Most reviewers appreciate the effectiveness of the proposed method. However, after the detailed author-reviewer discussion, Reviewer xKH7 and Reviewer Jv1W still have concerns about the motivation and superiority of the paper. Reviewer Jv1W further pointed out that the density-aware sampling techniques have already been published in some early papers, and he/she thinks that the performance is not strong enough considering that this is not a theoretical paper.

**Justification For Why Not Higher Score:**

N/A

**Justification For Why Not Lower Score:**

N/A

---

### Decision · Program_Chairs · 2024-01-16

Reject